# NK cells and CD38: Implication for (Immuno)Therapy in Plasma Cell Dyscrasias

**DOI:** 10.3390/cells9030768

**Published:** 2020-03-21

**Authors:** Renato Zambello, Gregorio Barilà, Sabrina Manni, Francesco Piazza, Gianpietro Semenzato

**Affiliations:** 1Department of Medicine (DIMED), Hematology and Clinical Immunology Section, University of Padova, 35128 Padova, Italy; r.zambello@unipd.it (R.Z.); gregorio.barila@gmail.com (G.B.); sabrina.manni@unipd.it (S.M.); francesco.piazza@unipd.it (F.P.); 2Veneto Institute of Molecular Medicine (VIMM), 35129 Padova, Italy

**Keywords:** CD38, NK cells, multiple myeloma

## Abstract

Immunotherapy represents a promising new avenue for the treatment of multiple myeloma (MM) patients, particularly with the availability of Monoclonal Antibodies (mAbs) as anti-CD38 Daratumumab and Isatuximab and anti-SLAM-F7 Elotuzumab. Although a clear NK activation has been demonstrated for Elotuzumab, the effect of anti-CD38 mAbs on NK system is controversial. As a matter of fact, an initial reduction of NK cells number characterizes Daratumumab therapy, limiting the potential role of this subset on myeloma immunotherapy. In this paper we discuss the role of NK cells along with anti-CD38 therapy and their implication in plasma cell dyscrasias, showing that mechanisms triggered by anti-CD38 mAbs ultimately lead to the activation of the immune system against myeloma cell growth.

## 1. Introduction

Natural killer cells are a group of innate lymphoid cells (ILCs) with strong cytotoxic function against stressed cells, such as virus-infected cells or tumor cells. They represent 5–15% of human peripheral blood mononuclear cells (PBMC) and tissue-resident NK cells can be found in the skin, spleen, liver, lungs, and other organs under physiological conditions [1]. NK cells in the blood appear as large lymphocytes with numerous cytoplasmic granules and can be distinguished from other lymphoid cells by the absence of T- and B-cell-specific markers, such as CD3 and CD19, and the presence of neural cell adhesion molecule (NCAM) CD56. Two main human NK cell subsets can be distinguished based on CD56 density on the cell surface: CD56^bright^ and CD56^dim^. CD56^bright^ NK cells are the major subset of NK cells in secondary lymphoid tissues and represent a less mature stage of NK cell differentiation, whereas CD56^dim^ cells represent the majority of NK population in the peripheral blood (80–95%) [2]. The downregulation of CD56 is associated with the acquisition of a high cytotoxic potential and this reflects the distinct physiological roles of the two NK cell subsets: CD56^bright^ population is specialized in the production of inflammatory cytokines and chemokines, while the cytotoxic function resides primarily in CD56^dim^ cells [3]. The different functions of CD56^bright^ and CD56^dim^ populations also reflect the presence of distinct NK receptors and other molecules on the surface of the two subsets including CD16, which is expressed on most CD56^dim^ cells and in a limited subset of CD56^bright^ cells.

### 1.1. Development and Maturation of NK Cells

Human NK cells develop primarily in the BM and, unlike T cells, do not require thymus for their maturation. However, subsets of NK cells have been shown to develop in secondary lymphoid organs, including lymph nodes and thymus, and in the liver [4,5]. NK cell development in the BM from the common lymphoid progenitor (CLP) proceeds through distinct maturation stages still not completely characterized based on sequential acquisition of NK cell-specific markers and functional competence. Expression of CD122 (IL-2Rβ) marks the irreversible commitment of CLPs into NK lineage, while the appearance of CD56 indicates a final transition from immature NK cells to mature NK cells, together with the expression of CD57 as a marker of terminal differentiation. Downregulation of CD56 expression from bright to dim levels marks the final differentiation stages and is associated with the appearance of CD16 receptor (FcγRIII). Several cytokines are essential to NK cell survival. In particular, IL-15 was shown to be crucial for the growth of NK cells and for the homeostasis and survival of peripheral NK cells. IL-2, IL-7 and IL-21 have important, albeit less characterized, roles in sustaining NK cell proliferation and survival, as well [6].

During their development, NK cells undergo an educational process involving the engagement of inhibitory killer immunoglobulin receptors (KIRs) with cognate MHC class I molecules. Inhibitory KIR expression during NK cell development is essential for the establishment of the “missing-self” recognition, a process by which NK cells preferentially recognize and kill cells that have lost the expression of self MHC class I molecules. The number of interactions between inhibitory receptors on developing NK cells and MHC class I molecules on stromal and hematopoietic cells in the bone marrow determines the degree of responsiveness of mature NK cells. In contrast, NK cells that lack inhibitory receptor expression during their development or are unable to interact with MHC class I molecules become hyporesponsive (anergic) cells [4]. This mechanism allows for the self-tolerance of NK cells towards self, healthy, MHC class I-expressing cells.

### 1.2. NK Cell Receptors

NK cell activity is regulated by the fine integration of signals coming from two distinct subsets of receptors on the cell surface: inhibitory and activating receptors. In contrast with TCR, NK receptors are germline-encoded and do not undergo somatic rearrangement during development. NK cells, indeed, are ready to fully respond to an infection or to the presence of malignant cells without a prior antigen-driven activation. Inhibitory receptors prevent the killing of target cells and they mainly bind MHC class I molecules leading to self-tolerance; loss of MHC class I expression is, instead, a mechanism adopted by virus-infected or tumor cells to avoid immune recognition by cytotoxic T lymphocytes (CTLs), and this leads to lower inhibitory signals in NK cells. Conversely, cellular stress induced by viral infections or tumor development promotes the upregulation of ligands on the stressed cells, which can be recognized by activating receptors [1]. The balance of signals from “stress-induced self” and “missing-self” determines whether an individual NK cell will be triggered to kill a target cell or not.

A large variety of activating receptors is present on mature NK cell surface, many of them being members of the killer immunoglobulin like receptor (KIR) family. Activating KIRs have short cytoplasmic tails, indicated by the “S” in KIR names as KIR3DS1 and KIR2DS1, and are non-covalently associated to DAP12 adapter molecule for downstream signaling. KIRs contain different numbers of immunoglobulin (Ig) domains on the extracellular side, reflected by the first number in the KIR designation (e.g., KIR3DS1 has three Ig domains, and KIR2DS1 has two Ig domains). The expression of the KIR repertoire is highly heterogeneous among different individuals, due to the difference in the expression of KIR molecules on individual NK cells as well as allelic variation in KIR genes [7]. A second important group of activating NK receptors belongs to the family of C-type lectins. A well-studied member of this family is NKG2D receptor, which binds MHC class I-related proteins, including MIC-A and MIC-B, found on virus-infected and tumor cells but not in normal cells [8]. NKG2D associates with the DAP10 signaling subunit and forms a hexameric complex composed of a single NKG2D homodimer along with two DAP10 homodimers [8]. In addition to NKG2D, two other members of C-type lectin family, NKG2C and NKG2E, act as activating receptors. They form heterodimers with the CD94 molecule and associate to the DAP12 adaptor molecule for downstream signaling activation. Natural cytotoxicity receptors (NCRs) are another immunoglobulin superfamily of activating receptors; human NK cells express three distinct types of NCRs, NKp46, NKp44 and NKp30, recognizing a wide variety of ligands on target cells, including bacterial and viral proteins and tumor-associated molecules [7]. NK cells are able to recognize infected cells that have been coated with antibody molecules through CD16 (FcγRIIIA), which is a low-affinity receptor for Fc region of IgG antibodies. CD16 engagement with its ligands generates activating signals in NK cells leading to the killing of the target cell through a process called antibody-dependent cell-mediated cytotoxicity (ADCC).

Adaptor molecules associated with stimulating receptors contain immunoreceptor tyrosine-based activation motifs (ITAMs) in their cytoplasmic domains, which are phosphorylated by tyrosine kinases of the Src family upon ligand–receptor complex formation. Phosphorylated DAP10 recruits and activates the p85α subunit of PI3K or Grb2 adaptor protein, whereas DAP12 recruits Zap70 and Syk tyrosine kinases to initiate downstream signaling, leading to the activation of mitogen-activated protein kinases (MAPKs), extracellular signal-regulated kinases (ERKs) and NF-kB pathways. The outcome of these signals results in a reorganization of actin cytoskeleton, allowing the release of cytolytic granules, and in the transcription of cytokine genes [1,6].

Inhibitory signals are triggered through immunoreceptor tyrosine-based inhibitory motifs (ITIMs) present in the cytoplasmic tails of inhibitory receptors. Upon ligand binding, ITIMs are phosphorylated and recruit tyrosine phosphatases, such as SHP-1 (Src homology containing tyrosine phosphatase-1) and SHP-2, which remove phosphate groups from several proteins, thus preventing the downstream signaling generated by NK activating receptors. The largest group of inhibitory receptors belongs to the same KIR family that includes activating receptors; inhibitory KIRs bind a variety of MHC class I molecules and have long ITIM-containing cytoplasmic tails. The presence of a long tail is indicated by the letter “L” in KIR names, as KIR3DL1 and KIR2DL1; KIRs with long cytoplasmic domains are all inhibitory, with the exception of KIR2DL4, which is an activating receptor through a complex downstream mechanism. C-type lectin family inhibitory receptors include CD94/NKG2A heterodimer, which recognizes the class I MHC molecules HLA-E [9]. Interestingly, HLA-E display peptides derived from other MHC class I molecules, rendering CD94/NKG2A a surveillance receptor for several distinct MHC I molecules.

### 1.3. NK Cell Effector Functions

Activating receptor-driven stimulation triggers the activation of NK cell effector mechanisms, including direct killing of the target cells and cytokine production, both being essential components of the innate immune response.

NK cell cytotoxic response involves three subsequent steps: 1) target cell recognition, 2) formation of an immunological synapse (IS) between NK cell and target cell, and 3) NK cell-induced target cell death. The primary mechanism of target cell killing involves the release of cytotoxic molecules such as perforin and granzymes contained in NK cell granules, through a degranulation process. Along with the formation of the IS, the NK cell reorganizes its actin cytoskeleton, leading to the polarization of microtubule organizing center (MTOC) and cytolytic granules towards the site of contact with the target cell, where granules are exocytosed and their content is released. Perforin polymerizes and forms pores in the target cell membrane, facilitating the delivery of granzymes into the cytoplasm. Granzymes are members of a serine protease family and they are capable of inducing apoptotic cell death by different pathways, including direct cleaving and activation of effector caspases-3 and -7, and Bak/Bax-driven release of cytochrome C from mitochondria. Granzymes also regulate the production of proinflammatory cytokines (IL-1β) by a mechanism dependent on caspase-1 [7].

Another process by which NK cells mediate killing of target cells involves the activation of death receptors of the tumor necrosis factor receptor (TNFR) superfamily that are expressed on the target cells. The two most prominent apoptosis-inducing TNFR family members are Fas (CD95) and TNF-related apoptosis-inducing ligand-receptor (TRAIL-R). Their respective ligands, Fas ligand (FasL) and TRAIL, can be present on NK cell surface or secreted by NK-derived exosomes, and they are bound by death receptors, promoting receptor oligomerization. Consequently, oligomerized death receptors recruit Fas-associated death domain (FADD) that, in turn, binds procaspase-8 allowing its activation and triggering apoptotic pathways [4]. Since NK cells are equipped with functional FcγRIII/CD16, the presence of antibody on a cell surface delivers a strong activating signal to NK cells to induce lysis of antibody-bound cells and antibody-dependent cell cytotoxicity.

The different NK cell effector mechanisms mediating the direct killing of the target cell are shown in Figure 1.

Activated NK cells modulate the immune response by their ability to produce pro-inflammatory cytokines, most notably Interferon-γ (IFN-γ), Tumor necrosis factor (TNF) and granulocyte/monocyte colony-stimulating factor (GM-CSF), which facilitate the activation of T cells and other innate immune mediators. Moreover, they release chemokines that can attract additional lymphocytes and myeloid cells to inflamed tissues [6]. INF-γ is one of the most potent effector cytokines secreted by NK cells: It plays a crucial role in antibacterial, antiviral and antitumor response, and it has been shown to induce the surface expression of death receptors on target cells [1].

Although NK cells do not utilize clonotypic receptors, such as the TCR, recent reports from several laboratories have identified relatively small populations of NK that exhibit immunological memory. The innate NK cell memory consists in a more rapid and robust NK cell response during secondary infections and it can be generated in response to certain viruses, as well as upon combined cytokine activation [2]. In humans, memory NK cells have been described in response to cytomegalovirus (CMV) infections, resulting in an increased frequency of CD94/NKG2C^+^ NK cells. This memory population expands during both an acute infection and a secondary challenge, and shows an increased INF-γ and TNF production in response to target cell stimulation [1]. In addition, cytokine-induced memory-like (CIML) NK cells have been reported to develop in response to IL-12, IL-15 and IL-18 combined pre-activation; CIML NK cells exhibit an increased production of INF-γ after a second stimulation with cytokines or in response to tumor target cells, and they are characterized by enhanced cytotoxicity against malignant cells [1,2].

## 2. NK Cells and Plasma Cells Dyscrasias

Immune dysfunction assumes a crucial role in the progression of precursor states: Monoclonal Gammopathy of Undetermined Significance (MGUS) and smouldering MM (sMM) into overt MM. Although myelomatous plasma cells often loose the HLA surface expression, they manage to escape from NK-mediated cell killing, suggesting that, in patients with MM, NK cell are dysfunctional [10].

Several studies evaluated NK cell counts in peripheral blood of MM and MGUS patients and even if they reported controversial results, most of them showed no significant changes [11,12,13]. Recently, we collected about 900 bone marrow samples of 715 patients with plasma cell dyscrasias and confirmed that no significant NK differences in percentages were found between MGUS, sMM and active MM cases [14], hinting that NK functional abilities rather than simply counts are impaired. As a matter of fact, progression of MGUS to MM is characterized by the reduction of cytotoxic properties and acquisition of an “exhausted” phenotype of NK cells, often due to changes in NK receptor repertoire, with reduction of NKG2D, NKp30 and DNAM-1 receptors. Moreover, increased PD-1 expression in NK cells and its bond to PD-L1 presented by MM cells, is responsible for immune response inhibition and MM progression [10,15].

Considering the importance of NK cell impairment in the disease progression, especially in the more advanced stages, several studies have evaluated the impact of specific anti-myeloma treatments on NK cell function [16,17,18,19,20,21,22]. Nevertheless, immunomodulatory drugs (IMiDs) like lenalidomide and pomalidomide exert at least some of their anti-myeloma effects increasing NK cell cytotoxicity [15]. Paiva and colleagues analyzed the immune status of high-risk sMM treated with lenalidomide and dexamethasone in the QUIREDEX trial. Even though no differences statistically significant were found in CD56^bright^ and CD56^dim^ NK cell counts before and after treatment towards healthy donor, lenalidomide exposure was able to induce a shift in NK cell phenotype to an activated status [18].

An increase in NK cell maturation and activation was observed also during lenalidomide maintenance in the IFM/DFCI 2009 trial, with an upregulation of CD16 and CD57 antigens and progressive expression of the NKG2C receptor. The authors, however, assumed that these changes were mostly due to immune reconstitution after autologous stem cell transplantation and CMV reactivation, since they persisted after lenalidomide interruption [19].

In conclusion, NK cells’ functional alterations parallel disease progression, thus implying that therapeutic strategies based on the NK effector functions are more likely to be effective when used in the early phases of the disease.

Recently, genetically engineered T lymphocytes to express synthetic chimeric antigen receptors (CARs) against specific antigens represent a novel intriguing therapy for MM patients [23,24]. Despite the impressive results, CAR-T therapy retains some concerns regarding toxicities (i.e., cytokine release syndrome, off target toxicities). To overcome these issues, genetic modification of innate immune system cells like NK cells with CAR can represent an attractive alternative target. As a matter of fact, NKG2D-CAR-transduced NK cells demonstrated higher killing activity and efficacy in targeting Multiple Myeloma cells towards NKG2D-CAR-transduced T cells [25].

## 3. CD38 Expression and Function in Immune (NK) Cells

CD38 is a type II 45 kDa glycoprotein usually present in the cellular surface membrane. CD38 has multifaceted roles since it possesses properties of an activation marker, an adhesion molecule interacting with endothelial CD31 and ecto-enzymatic activity. CD38 is also an intracellular signaling protein. Studies on the relationship between the CD38 molecule and immune cells started years ago. Indeed, since its first identification and characterization, the expression of the CD38 molecule has been thoroughly investigated in leukocytes [26,27]. The early phases of hemopoietic differentiation and commitment towards the common lymphoid progenitor stage require the expression of CD38 together with other specific markers [28]. Subsequently, in more mature stages of development of immuno-hematopoietic cells, it has been reported that there are some differences in the pattern of CD38 expression. NK cells predominantly express CD38 in a constitutive manner [29], whereas CD34 precursors T and B lymphocytes show a pattern of expression whereby the CD38 molecule is upregulated upon activation or different functional status [30]. Subsequent research discovered that CD38 is physically/functionally clustered with critical T and B-cell membrane molecules, such as the TCR, BCR, CD19, and in NK-cells with FcγRIII/CD16 [30,31,32]. The contact of CD38 with these pivotal molecules for lymphocyte function is needed for signal transduction and generation of downstream processes, such as initiation of specific transcriptional programs, secretion of cytokines and activation of lymphocyte effector functions [31,32].

In activated NK cells, CD38 engagement by a monoclonal antibody evoked a cytotoxic response with the release of granzymes and cytokines. It was shown that IL-2-dependent protein expression was important to link CD38 to the NK effector response [33]. Also in NK cells, CD38 association with CD16 was reported to be important for CD16-expressing NK cells to acquire the effector cytotoxic phenotype [34] (Figure 2A). The association between CD38 and CD16 was found to be instrumental for the CD38-mediated activation of signaling events: the stimulation of CD38 with agonistic antibodies led to the interaction of CD38 and CD16. This interaction was described to be followed by the initiation of Ca^++^ flux, tyrosine phosphorylation of ZAP70, activation of MAPK, secretion of IFNγ, and the establishment of a cytotoxic response [34]. Interestingly, it has been shown that CD38 is involved in the functional activity of a NK subset characterized by the phenotype CD56^bright^ CD16^-^, which takes part in the antitumoral immune response, mainly through the release of regulative cytokines [35]. Intriguingly, CD38-mediated production of adenosine (ADO) by CD56^bright^ CD16^-^ NK cells accounts for CD4+ T cell inhibition and could be reverted by blocking CD38 with antagonist antibodies [36] (Figure 2B). Overall, these data suggest that CD38 is a peculiar signaling molecule with functional relevance for B, T lymphocytes, and NK cells.

## 4. Effects of CD38-Directed Immunotherapy on Immune Cell Levels and Function

By virtue of its expression and role in immune cells, the therapeutic targeting of CD38 is likely to display effects on the immune response. The first strategy to target myeloma CD38 has relied upon monoclonal antibodies (mAbs), which have been developed since the early 90s.

Stevenson et al. designed a chimeric mAb from the human CD38-specific mouse Ab OKT10, made by the mouse Fab portion of OKT10 (specific for CD38) fused by a thioester bond to a human IgG1 Fc chain. This chimeric mAb displayed ADCC activity with human PBMCs at low concentrations, apparently without affecting the viability and function of NK cells, granulocytes and mononuclear cells [37].

Currently available mAbs for clinical use are Daratumumab and Isatuximab. Daratumumab is a human IgG1κ mAb produced in Chinese hamster ovary (CHO) cells by recombinant DNA technology [38]. The efficacy of Daratumumab against MM has been extensively confirmed in the clinical setting, both in the relapsed/refractory as well as in the first line treatment [39,40,41]. The mechanism of action of Daratumumab relies mostly on ADCC and CDC. In vitro MM cell killing is elicited even in the presence of protective stromal cells [38]. Other studies have shown that Daratumumab can elicit direct cell killing through apoptosis if cross-linked by a secondary antibody or by FCγR [42].

Anti-CD38 mAbs have been described to interfere with immune cell populations (Figure 3). Daratumumab may cause a depletion of CD38^bright^ T-reg suppressor cells, while increasing the frequency of T-helper and T-cytotoxic cells, their clonality and their effector functions [43].

It has also been demonstrated that depletion of CD38^high^ NK cells occurs in vitro and in vivo in patients treated with Daratumumab [44]. On the other hand, the residual CD38^low/-^ NK population might display a high proliferative potential and a fully functional activity and, therefore, it has been proposed that it could be employed therapeutically [44]. Furthermore, a study has evaluated how to improve NK-dependent anti-MM Daratumumab-triggered ADCC [45]. The NK-dependent cytotoxicity against MM cells was evaluated in stressing conditions mimicking the tumor microenvironment, which was reported to inhibit NK cells. Daratumumab was demonstrated to elicit ADCC in the presence of tumor microenvironmental stressing conditions. The analysis of the anti-MM response of KIR ligand HLA-I matched versus HLA-I mismatched NK cells revealed that Daratumumab could elicit a stronger activation for KIR-ligand mismatched NK cells under stressing conditions [45]. Other studies have analyzed the pharmacodynamic effects of Daratumumab on NK cells as well as on the consequences of NK cells levels on Daratumumab efficacy and toxicity in early phase I and II clinical studies [46]. It has been observed that NK cells are rapidly cleared off after Daratumumab therapy and recover upon termination of therapy. These studies have also shown that on one hand the residual NK cells in the peripheral blood are fully capable of inducing ADCC against MM cells and, on the other hand, that there is no increased incidence of grade 3 or more side effects (especially infections) according to NK cell number reductions [46].

The other clinical grade anti-CD38 mAb, Isatuximab, is a human IgG1κ mAb under early clinical trials, in which it has shown meaningful anti-tumor activity [47,48]. Compared to Daratumumab, Isatuximab displays a stronger action of direct cell killing independent on cross-linking. These effects are perhaps mediated by different mechanisms, such as activation of caspases 7 and 8, lysosome permeabilization, and upregulation of reactive oxygen species (ROS) [49]. Isatuximab also activates ADCC, CDC and ADCP [50]. Isatuximab alone or in combination with the IMiDs pomalidomide and dexamethasone may modulate T-reg levels and function depending on the surface expression of CD38. Indeed, it was demonstrated that CD38 is expressed at higher levels in T-reg as compared to conventional T (Tcons) cells in MM patients. Isatuximab was shown to cause a preferential depletion of T-reg, relieving Tcons from T-reg suppression, and this effect is enhanced by IMiDs [51]. T-regs depletion is caused by Isatuximab-induced apoptosis and inhibition of cell proliferation. Isatuximab-mediated T-reg inhibition contributes to elicit CD8+ T cell and even NK cell-induced MM cell killing, which is further enhanced by IMiDs [51]. Even if a certain extent of NK cell depletion is reported upon Isatuximab exposure [49], CD8+ T and NK cells are believed to be substantially spared from Isatuximab cytotoxicity, likely due to the preferential action on CD38-high expressing T-regs.

Altogether, these data suggest that therapeutic targeting of CD38 with mAbs in MM can cause advantageous functional consequences on the anti-MM immune surveillance mechanisms.

## 5. Conclusions 

NK cells play a crucial role in immunotherapy against myeloma cells. Anti-CD38 mAb therapy has been reported to impact on this immune compartment by reducing the number of circulating NK cells. This finding indicates the possibility of a negative impact on NK cell function mediated by anti-CD38 mAbs. However, with all the mechanisms triggered by anti-CD38 mAbs, taken together, the final result ultimately leads to the activation of the whole immune system. Considering the relevance of NK cells on the mechanisms controlling myeloma plasma-cell growth, the clinical impact of this final effect remains obvious. Furthermore, the more frequent combination use of anti-CD38 mAbs, in particular with IMiDs, is likely to trigger a powerful response of all the immune competent apparatus against Myeloma cells with unexpected results, thus paving the way for a putative cure of the disease.

## Figures and Tables

**Figure 1 cells-09-00768-f001:**
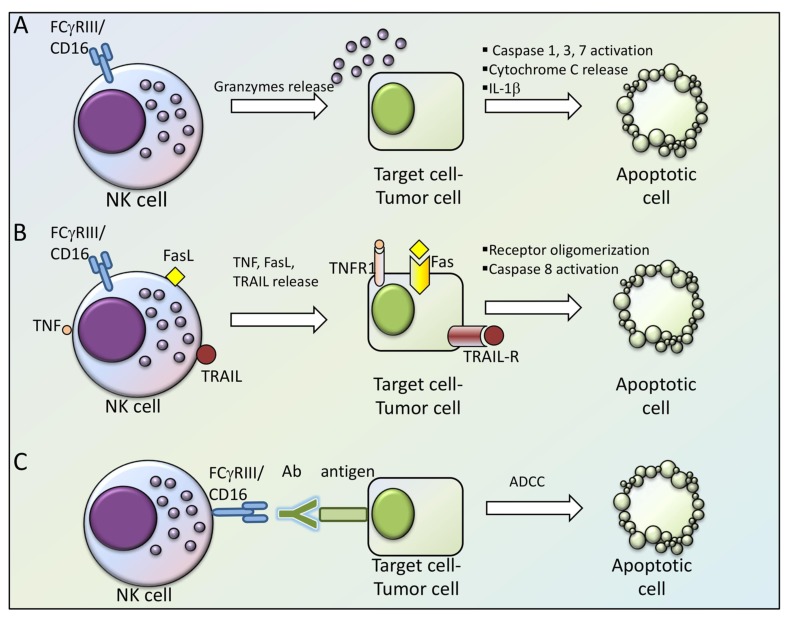
Mechanisms accounting for cytotoxic activity of NK cells. (**A**) FCγRIII/CD16-expressing NK cells release cytotoxic molecules such as perforin and granzymes contained in NK cell granules, which enter in the target cell/tumor cell leading to its apoptosis through caspase-1,3,7 activation, cytochrome C release and IL-1β expression. (**B**) TNFR family ligands (TNF, FasL or TRAIL) are expressed on, or secreted by NK cells and by binding to their receptors (TNFR1, Fas or TRAIL-R) on the target/tumor cell, induce its apoptosis through Caspase 8 activation. (**C**) The binding of FCγRIII/CD16 to an antibody (Ab) recognizing a target antigen on the tumor cell, induces its apopotosis through ADCC. Abbreviations used: NK= Natural Killer; ADCC (antibody dependent cell cytotoxicity); TNF= tumor necrosis factor; TNFR= tumor necrosis factor receptor; FasL= Fas ligand; TRAIL= TNF-related apoptosis-inducing ligand; TRAIL-R= TNF-related apoptosis-inducing ligand-receptor; Ab= antibody; IL-1β= Interleukin 1β.

**Figure 2 cells-09-00768-f002:**
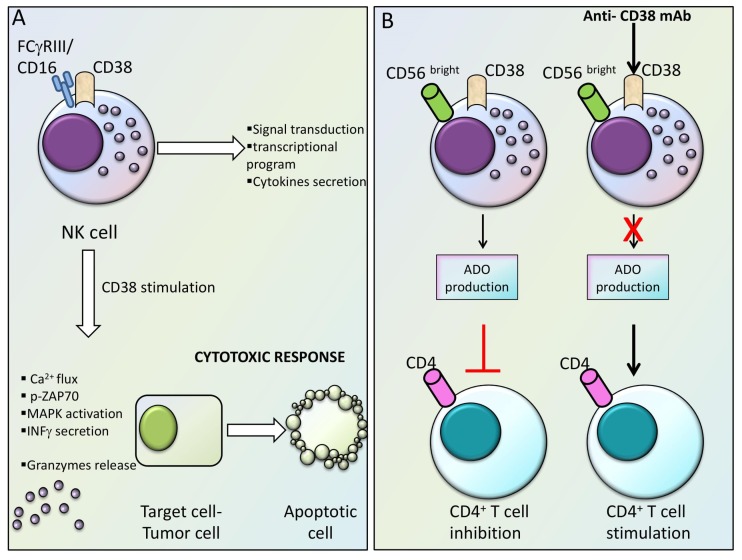
CD38 expression and function in NK cells. (**A**) Role of CD38 in the activation of NK cells. (**B**) Role of CD38 in the functional activity of CD56^bright^/CD16^-^ NK cells.

**Figure 3 cells-09-00768-f003:**
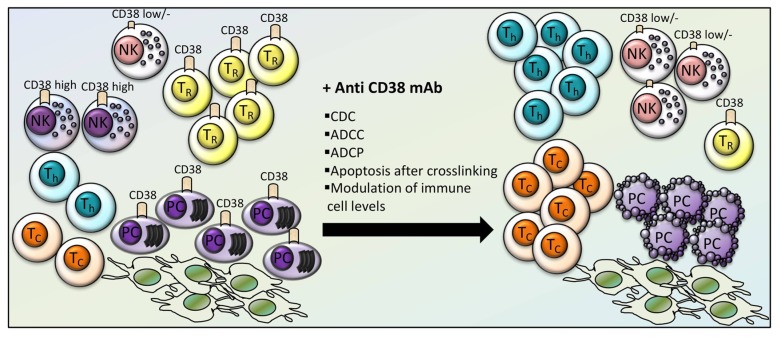
Effects of anti-CD38 therapy on immune cell levels in MM. Anti-CD38 monoclonal antibodies (anti-CD38 mAb) have shown a cytotoxic effect on MM plasma cells (PC) even in the presence of protective bone marrow stromal cells, through CDC (complement-dependent cytotoxicity), ADCC (antibody-dependent cellular cytotoxicity), ADCP (antibody-dependent cellular phagocytosis), and apoptosis after cross-linking. Anti-CD38 mAb increased the frequency, clonality and function of T-helper (T_h_) and T-cytotoxic (T_C_) cells, and CD38^low^ NK cells, while depleting CD38^high^ NK and CD38-expressing T-reg (T_R_) suppressor cells.

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
