# Peer review of "NK cells and CD38: Implication for (Immuno)Therapy in Plasma Cell Dyscrasias"

_cells, 2020, doi:10.3390/cells9030768_

Round 1

Reviewer 1 Report

In their review titled “NK cells and CD38: implication for (immuno)therapy in plasma cell discrasias” Zambello et al. describe the role the NK cell-mediated immune system in CD38-targeting antibody treatments for plasma cell dyscrasias (and not discrasias!!). Overall, the review is well written and provides some significant additional information over what has already been published.

There is one issue that keeps the manuscript from being publishable in its current form. The problem is that it remains somewhat unclear to what degree the findings they describe in their review are of clinical relevance. To some degree they fail to explain why this topic needs to be addressed.

Author Response

We thank for the Reviewer’s comment. Concerning NK cells and CD38 expression on their cell surface and their recognized role in counteracting myeloma cell growth, we believe that the data we provide support the evidence that the immunotherapy with anti-CD38 (whatever) mab, (which is characterized by a reduction of NK cells) ultimately results in the activation of the system. Accordingly, these concepts fully address the topic of this special issue (CD38 and Disease: A Bi-Directional Cross-Talk between Pathology and Physiology) and also imply a relevant clinical impact in the comprehension of the mechanisms of CD38 mAb immunotherapy. A sentence summarizing this concept has been added in the “Abstract” and in the “Conclusions” sections. Furthermore, a paragraph has been included (pag. 5) on CAR-NK strategy in the scenario of new therapies for Myeloma that further emphasizes the potential role of these cells. All changes are marked in yellow.

Reviewer 2 Report

This is a well-described review article. Other than minor grammatical errors, there is no major issues found.  

Author Response

We thank the Reviewer for comments. Grammatical errors have been corrected.

Reviewer 3 Report

The manuscript is of quite interest and is easy to comprehend. There is no criticism or comment about the paper. There are a couple misspelling.

Author Response

We thank the Reviewer for the comments. Spelling errors have been corrected.

Reviewer 4 Report

Strength

  1. Cancer immunotherapy is a hot topic and must attract a lot of reads.
  2. The manuscript is well written and easy to understand.

Weakness

  1. The authors should provide more references for several sentences. Line 79, 81, 109, 180, etc.
  2. Line 271: Add (TR) after T-reg.
  3. Line 272: Subclass is missing “IgGk mAb”
  4. Should use same format throughout the manuscript. For example, “Anti-CD38” is written like “AntiCD38” or “Anti CD38”. “IMIDs” is written like “IMiDs” or “IMIds”.

Author Response

We thank the Reviewer for the comments. New references have been added. Other suggestions made by the Reviewer have been also accepted and corrections have been made accordingly. All changes are marked in yellow.

Round 2

Reviewer 1 Report

I have no further comments.